# The Role of Antibiotic Prophylaxis in Anastomotic Leak Prevention during Elective Colorectal Surgery: Systematic Review and Meta-Analysis of Randomized Controlled Trials

**DOI:** 10.3390/antibiotics12020397

**Published:** 2023-02-16

**Authors:** Lidia Castagneto-Gissey, Maria Francesca Russo, James Casella-Mariolo, Angelo Serao, Rosa Marcellinaro, Vito D’Andrea, Massimo Carlini, Giovanni Casella

**Affiliations:** 1Department of Surgery, Sapienza University of Rome, Viale Regina Elena, 324, 00161 Rome, Italy; 2Department of General and Emergency Surgery, Ospedale dei Castelli (NOC), ASL Roma 6, 00072 Rome, Italy; 3Department of General Surgery, S. Eugenio Hospital, 00144 Rome, Italy

**Keywords:** colorectal surgery, anastomotic leak, surgical site infection, antibiotics, prophylaxis

## Abstract

*Introduction*: Despite several perioperative care advancements and innovations in surgical procedures and technologies, the incidence rate of anastomotic leaks (ALs) after colorectal surgery has not substantially decreased. Gut microbiota can play a critical role in the healing process of anastomotic tissue and alterations in its composition may be largely to blame for anastomotic insufficiency. The use of specific antibiotics for preoperative large bowel decontamination could significantly influence the rate of ALs. The aim of this study was to systematically assess the various antibiotic prophylactic regimen strategies for primary prevention of ALs during colorectal surgery, in view of the available evidence. *Methods*: A systematic review of the literature was conducted, and randomized clinical trials (RCTs) analyzing prophylactic antibiotic bowel preparation in colorectal surgery were included. PubMed, Embase, the Web of Science Core Collection, and the Cochrane Central Register of Controlled Trials were searched from inception through to 30 November 2022. The methodological quality of the included trials was evaluated. The primary outcome was AL rate; secondary outcomes were superficial/deep surgical site infections (SSIs). The PRISMA guidelines were used to carry out the present systematic review. *Results:* Thirteen RCTs published between 1977 and 2022, with a total of 4334 patients were included in the meta-analysis. Antibiotic prophylaxis was administered orally in 11/13 studies and intravenously in 2 studies. Patients randomly assigned to antibiotic prophylaxis, regardless of the regimen, had a reduced risk of ALs (*p* = 0.003) compared to mechanical bowel preparation (MBP) alone. The use of antibiotic prophylaxis was also more effective in significantly reducing SSIs (*p* < 0.001). *Conclusions:* The evidence points to an advantage of oral antibiotic prophylaxis in terms of AL rate, a significant contributor to perioperative morbidity, mortality, and rising healthcare expenditures. In light of such results, the use of antibiotic prophylaxis should be strongly encouraged prior to colorectal surgery.

## 1. Introduction

Despite several perioperative care advancements and innovations in surgical procedures and technologies which have been developed over the past few decades, the incidence rate of anastomotic leaks (ALs) after colorectal surgery has not substantially decreased. ALs are currently reported to develop in 6–8% following colonic resections and between 7 and 20% after rectal surgery [1]. When ALs occur, the risk of postoperative mortality, length of hospital stay, cancer recurrence, permanent stomas, and total expenditures rise significantly [2,3]. The patient’s characteristics, cancer stage, and surgical technique are just a few of the many identified risk factors for anastomotic failure.

The hypothesis that gut microbiota can play a critical role in the healing process of anastomotic tissue and that alterations in its composition may be largely to blame for anastomotic insufficiency has been proposed by some authors [4,5,6]. In the endocrine, neurological, and metabolic systems, it is well established that microbiota and the host have a mutualistic interaction. The primary functions of gut microbiota are to maintain healthy bowel function, protect the intestinal barrier’s integrity, support mucosal immunity, protect the mucosa from infections, and produce bioactive substances [7]. Therefore, it is essential to protect the microbiota’s biodiversity in order to retain these functions.

In this regard, it has been previously shown that any surgical operation performed on the gut can significantly alter this biodiversity, which may affect how the anastomosis heals. In a mouse model, Shogan et al. found how colonic resection followed by the creation of an anastomosis caused significant alterations in the gut microbiota’s composition [6]. On the other hand, Cohn et al. successfully demonstrated in an animal model, how experimentally inducing peri-anastomotic colonic ischemia and administering intraluminal antibiotics directly at the level of the anastomosis could allow correct healing of the bowel tissue despite local ischemia, compared to the control group who did not receive any antibiotics, where complete disruption of the anastomosis occurred [8].

In consideration of the fundamental role played by the gut microbiota in anastomotic healing, the use of specific antibiotics for preoperative large bowel decontamination could significantly influence the rate of ALs. The aim of this study was to systematically assess the various antibiotic prophylactic regimen strategies and their role in preventing ALs during colorectal surgery, in view of the available evidence.

## 2. Results

A total of 2151 studies were found in the electronic search. After reviewing titles and abstracts, 1828 studies were excluded, while 323 were screened. Out of these, 285 did not have anastomotic leakage as an endpoint. No eligible trials were found prior to 1973, and the first trial included in the meta-analysis was from 1977. The remaining 38 articles were analyzed and 25 were excluded because they did not compare antibiotic use with a placebo control group. Thus, 13 articles were included in the final analysis (Figure 1).

### 2.1. Methodological Quality Assessment

Regarding RCTs’ methodological quality, all studies reported using the random sequence generation methods and were double-blinded in eight studies, single-blinded in four studies, and open-label in one study. Methodological quality evaluated using Jadad’s validated scale revealed just one study with a score of 4 (7.7%); five studies obtained a score of 3 (38.5%); six studies (46.1%) obtained a score of 2; and one study received a score of 1 (7.7%) (Table 1). The overall risk of bias judgement according to the Cochrane risk-of-bias tool for randomized trials (RoB 2) was ‘high’ for 9 (69.2%) studies and with ‘some concerns’ for 4 (30.8%) trials (Table 2). The biggest risk of bias in this review derives from the selection of the reported result. The outcome domain anastomotic leak is not univocally measured and is mainly based on clinical judgement and/or radiologic exams, which may affect this outcome measure and increases the risk of bias.

**Table 1 antibiotics-12-00397-t001:** Methodological quality assessment according to Jadad scores.

Author	Jadad Scores for RCTs
Hojer et al. [9]	2
Matheson et al. [10]	2
Bartlett et al. [11]	1
Ishida et al. [12]	2
Sato et al. [13]	2
Sadahiro et al. [14]	2
Hjalmarsson et al. [15]	3
Anjum et al. [16]	3
Abis et al. [17]	3
Koskenvuo et al. [18]	3
Mulder et al. [19]	2
Papp et al. [20]	3
Futier et al. [21]	4

The *p* values for Egger’s and Begg’s tests for anastomotic leak were *p* = 0.0450 and *p* = 0.1127, respectively. Furthermore, the *p* value for Egger’s and Begg’s tests for surgical site infection was *p* = 0.0468 and *p* = 0.1795 (Figure 2 and Figure 3).

### 2.2. Primary Outcomes

A total of 4334 patients participated in the selected studies. All patients received mechanical bowel preparation (MBP). Two thousand forty-one patients received antibiotic prophylaxis, whereas 1943 received only MBP.

Table 3 shows the characterization of studies regarding the interventions investigated.

In 11 out of 13 studies, antibiotics were administered orally, while in the remaining two studies, patients received intravenous antibiotic prophylaxis (Table 4).

The individual and pooled odds ratio (OR) and risk differences of ALs are shown in Figure 4. The overall analyses indicated that patients randomly assigned to the antibiotic prophylaxis, regardless of the regimen or route of administration, had reduced risk of ALs (*p* = 0.003), compared with participants receiving MBP alone. Indeed, the meta-analysis average effect, represented by the center of the diamond, is located at the left of the vertical line center for both endpoints, favoring intervention (Figure 4).

### 2.3. Secondary Outcomes

Throughout the studies, the definition of organ-space infection varied. Particularly, just one study made a distinction between organ-space infections, with most studies describing either radiological or clinical signs of SSIs.

Individual and pooled ORs and risk differences for the secondary analyses of SSIs are shown in Figure 5. The analysis of included RCTs showed that antibiotic prophylaxis was more effective, with respect to MBP alone, in significantly reducing SSIs (*p* < 0.001).

## 3. Discussion

Anastomotic leaks are a significant postoperative complication following surgery for colorectal disease and have a major impact on patient morbidity and mortality. The RCTs included in the present meta-analysis clearly highlight the beneficial effect of antibiotic prophylaxis, regardless of the type of regimen used, on the rate of both ALs and SSIs (*p* = 0.003 and *p* < 0.001, respectively) compared to MBP alone. In light of such results, the use of antibiotic prophylaxis prior to colorectal surgery should be strongly encouraged.

From the studies retrieved in the present systematic review, there was insufficient data from each different combination of an oral antibiotic agent, dose, time, and parenteral antibiotic details to conduct a meaningful analysis between subgroups. According to the patient’s preparation regimen, some contained only one preoperative dosage of an oral antibiotic or used different parenteral antibiotic regimens, which could be significantly biased.

The numerous variables, including differences in the type of antibiotic used, dosing methods, and route of administration, made it difficult to identify which of these three variables may have caused any discrepancy in outcomes.

Most studies used antibiotic combinations without comparing them to recognized prophylaxis approaches or offering a logical prophylaxis strategy based on what is known about colonic bacterial flora and the characteristics of postoperative ALs and SSIs in colorectal surgery.

Although gold-standard antibiotic regimens have been proposed by various scientific societies, with the goal of including the gold standard as the appropriate benchmark from which to judge the new antibiotic in all future investigations, only a few of the eligible studies actually implemented these recommended antibiotics.

Evidence from the review’s analysis suggests that the chosen antibiotic should cover both aerobic and anaerobic microorganisms. These results provide evidence that treating both types of bacteria, whether with a single agent or in a combination therapy, is more effective in reducing SSIs and ALs than treating only aerobic or only anaerobic bacteria.

Further investigation is necessary to determine the best timing and duration for dosing, as well as the incidence of longer-lasting negative effects, including Clostridium difficile pseudomembranous colitis.

The use of oral antibiotics with or without mechanical bowel preparation was initially investigated in the 1970s but later fell out of favor, reemerging only recently as a viable intervention given the high rate of superficial and organ/space SSIs, as well as ALs following colorectal surgery.

In response to specific conditions, such as large tumors, substantial blood loss, prolonged surgical procedures, and ischemia, the host produces local and systemic inflammatory signals during anastomotic construction. Bacteria respond to this altered environment by genic shifting and/or activation, resulting in increased virulence. If they are more prevalent, these pathogenic activated elements can bind to the anastomotic tissues, evade the immune system, and activate collagenase genes, intensifying the inflammatory response in the tissue. In addition to degrading collagen I, bacterial collagenases can also cause local tissue matrix metalloproteinases-9 (MMP-9) to degrade collagen IV. The anastomotic tissue can rupture as a result of this process [7].

Prior to surgery, identifying risk variables is a crucial step that enables the management of modifiable factors and surgical technique adaptation. The best course of action for this continues to be AL prevention, despite the lack of success of endorsed preventive programs to date.

Bowel regimes before elective colorectal surgery are rarely implemented across many institutions and are still primarily dependent on the surgeons’ preference. Oral antibiotic prophylaxis is typically not required, and even the type of mechanical bowel preparation varies from center to center.

In fact, the use of antibiotic prophylaxis is still not broadly regarded as a standard of care, despite research released in 1981 that already demonstrated its value in reducing infection rates and overall mortality, as well as establishing the need for no more placebo or control trials [22]. A Cochrane Review that showed a substantial decrease in surgical site infections in patients receiving parenteral antibiotic prophylaxis compared to those receiving no antibiotics or a placebo (RR 0.34, 95% CI 0.28–0.41, *p* = 0.0001) presented conclusive evidence in favor of this claim [23].

The 2019 clinical practice Guidelines of the American Society of Colon and Rectal Surgeons strongly advise mechanical bowel preparation paired with preoperative oral antibiotics [24]. This association appears to improve short-term oncological outcomes and lowers the frequency of surgical site infections, anastomotic leakage, and postoperative ileus [25].

Eight cohort studies and 23 RCTs published between 1980 and 2015 were included in a recent meta-analysis [26]. However, the authors also included various cohort studies emerging from the American College of Surgeons National Surgical Quality Improvement Program (ACS-NSQIP) database, which most likely implies duplicate reporting of the same patient information. According to this study, surgical site infection rates were significantly lower in cohort study participants who received mechanical bowel preparation, oral antibiotics, and IV antibiotics compared to those who received only mechanical bowel preparation and IV antibiotics (RR 0.48, 95% CI 0.44–0.52, *p* = 0.00001). In another recent meta-analysis, Bellows et al. compared the effectiveness of IV antibiotics alone against IV antibiotics in combination with oral non-absorbable antibiotics before colorectal surgery while concentrating on surgical site infections [27]. This study comprised 16 RCTs that were published between 1980 and 2011, with a total of 2669 patients, and all the trials had mechanical bowel preparation as part of the protocol. According to this meta-analysis, there was no significant difference in anastomotic leak rates between oral and IV antibiotics compared with IV antibiotics alone (RR 0.63, 95% CI 0.28–1.41, *p* = 0.3), although there was a substantial decrease in superficial wound infection rates.

Over the past 50 years, there have been great controversies surrounding the use of antibiotic prophylaxis with or without mechanical bowel preparation. The composition of gut microbiota is altered by mechanical bowel preparation alone. In particular, mechanical bowel preparation raises intestinal pH, which promotes Proteobacteria and Enterobacteriaceae expansion and a decrease in Lactobacillaceae, changing the ratio of Gram+ to Gram- bacteria. When mechanical bowel preparation is administered in two or more doses as opposed to one, these side effects are not severe and last for a shorter amount of time [28,29].

Although some recent meta-analyses concluded that mechanical bowel preparation alone cannot lower the risk of ALs or intra-abdominal abscesses following colorectal surgery [30,31], the beneficial association between mechanical bowel preparation and oral antibiotics has been noted since the 1970s in lowering surgical complications, such as surgical site infections [30,31,32].

There are currently only a limited number of studies in the literature that assess how well a combination of mechanical bowel preparation, oral antibiotics, and probiotics prepare patients for colorectal resections. Nevertheless, some studies have indicated that using Bifidobacteria postoperatively may help maintain an optimal microbiota balance. The use of probiotics lessens and modulates the inflammatory response, enhances healing, and improves the composition of fecal microbiota in patients undergoing colorectal surgery [32,33]. Probiotic use before surgery is associated with a lower occurrence of postoperative complications, primarily ALs and infections, promoting a healthy recovery [34]. This raises the prospect that, in the event of anastomotic leakage, a favorable microbiota, already present at baseline or orally/locally administered may result in less severe sepsis and peritonitis.

Although antibiotic prophylaxis has been widely implemented in clinical practice, the underlying mechanisms for its effectiveness in reducing the rates of such dreaded complications are not fully understood. For this reason, it appears difficult to comment on the mechanistic benefit of a prophylactic regimen due to the numerous factors which need yet to be explored.

### Study Limitations

Some limitations must be acknowledged in the present study. The most important factor contributing to clinical variability is the diversity of preoperative antibiotic prophylaxis regimens. Second, given the limited data and preliminary results, it is challenging to provide comprehensive guidance on the best prophylactic probiotic and antibiotic regimens for use in clinical practice. Third, the results were probably impaired by additional biases (mostly small trial bias), and only a few trials were sufficiently powered to address this problem. Finally, those studies not reporting anastomotic leaks either as a primary or secondary outcome were excluded from the present review as we considered this an incomplete outcome data bias.

## 4. Materials and Methods

### 4.1. Search Strategy and Selection of Trials

This study provides a systematic review and meta-analysis of previously published data emerging from randomized controlled trials, which was carried out according to the Preferred Reporting Items for Systematic Reviews and Meta-Analyses (PRISMA Statement) criteria [35] (PROSPERO Registration Number: CRD42023396144). The PICO strategy was used to formulate the guiding question: “What are the effects of antibiotics on anastomotic leakage and superficial skin infections during surgery for colorectal cancer?” [36]. The search was performed using the following electronic databases without any year restriction, from inception through to 30 September 2022: PubMed, Embase, the Web of Science Core Collection, and the Cochrane Central Register of Controlled Trials. All abstracts in the English language were screened for applicability. A manual search using the following keywords extracted from the Medical Subjects Heading (MeSH) was made: ‘colorectal surgery’ AND ‘antibiotics’ AND ‘antibiotic prophylaxis’ AND ‘leaks’ AND ‘dehiscence’ AND ‘surgical site infection’ AND ‘ssi’ AND ‘complications’ AND ‘randomized controlled trial’.

The eligibility criteria for the selection of articles, according to the PICO strategy were: RCTs with adults aged 18 years or over (population); use of preoperative antibiotics (intervention); comparison with no treatment (comparison); and incidence of anastomotic leakage and superficial skin infection (outcomes). The studies excluded were those not written in English, or that did not provide the full online abstract.

Two independent reviewers (LCG, MFR) screened and selected the trials to be included in the review. Conflicts were handled by consensus, and an adjudicator (JCM) was consulted when necessary. Only studies that were fully available and designed as randomized controlled trials evaluating the effects of different strategies of antibiotic prophylaxis before elective colorectal surgery and assessing postoperative complications (i.e., anastomotic leaks and surgical site infections) were included.

### 4.2. Critical Assessment of Trials and Collection of Data

Two independent reviewers evaluated the methodological quality of eligible trials using a validated scale [37]; in the event of a disagreement, the final score was decided by consensus. The quality scale included three components: (1) double-blinding, (2) randomization, and (3) dropouts. The scale ranged from 0 to 5, with a score of 0–2 for randomization, a score of 0–2 for blinding, and a score of 1 for dropouts. When the score is ≤2, a trial’s methodological quality is deemed poor/inappropriate according to Moher et al.

The two reviewers independently gathered data, which they then compared and cross-checked. Missing data was sought on the journal’s database and included if present. All studies with missing text or with insufficiently reported data were excluded.

Version 2 of the Cochrane risk-of-bias tool for randomized trials (RoB 2) was used to assess the risk of bias.

The following data were retrieved for each RCT: sample size, type/modality of randomization, blinding, dropouts, type of antibiotic prophylaxis regimen, use of mechanical bowel preparation, anastomotic leaks, surgical site infections, other postoperative complications, and mortality.

### 4.3. Outcome Measures

Outcomes were evaluated on the basis of intention-to-treat. The primary outcome was the rate of anastomotic leaks, while secondary outcomes included superficial and deep organ/space surgical site infections (wound infection, wound dehiscence, pelvic abscess, peritonitis).

### 4.4. Risk of Bias Assessment

The Cochrane risk of bias tool was used to assess RCTs. We evaluated randomization sequence generation, allocation concealment, blinding, completeness of outcome data, and selective reporting for each RCT. Concealment of allocation was considered adequate if the randomized method described in the text did not allow the investigators and the participants to know or influence the intervention group before the randomized allocation. Publication bias was evaluated using Begg’s and Egger’s tests [38,39].

### 4.5. Statistical Analysis

Data analyses were carried out using MedCalc v20.211 statistical software [40]. Odds ratios were selected to describe the ratio of odds in the treatment group to the odds of the control group. The heterogeneity among the studies was checked using Cochrane’s Q [41] and the I^2^ statistical tests [42,43]. The model of random effects was adopted for the analysis.

## 5. Conclusions

The current systematic review, which is the broadest and most comprehensive one to date assessing the role of different regimens of antibiotic prophylaxis in preventing ALs after colorectal surgery, suggests that oral antibiotic bowel decontamination, whether used alone or in conjunction with mechanical bowel preparation, may have a considerable impact on the reduction in postoperative complications. When just RCT-based evidence is taken into account, it tends to attenuate the considerable positive impact of antibiotics suggested by large retrospective cohort and database studies. The evidence, however, points to an advantage of oral antibiotic bowel preparation both in terms of SSI and AL rates, significant contributors to perioperative morbidity, and rising healthcare expenditures. In light of our results, the use of antibiotic prophylaxis prior to colorectal surgery should be strongly encouraged. Nevertheless, before making more firm recommendations, further high-quality evidence is needed to distinguish between the advantages of the different antibiotic regimens and whether this is more beneficial alone or combined with mechanical bowel preparation in this setting.

## Figures and Tables

**Figure 1 antibiotics-12-00397-f001:**
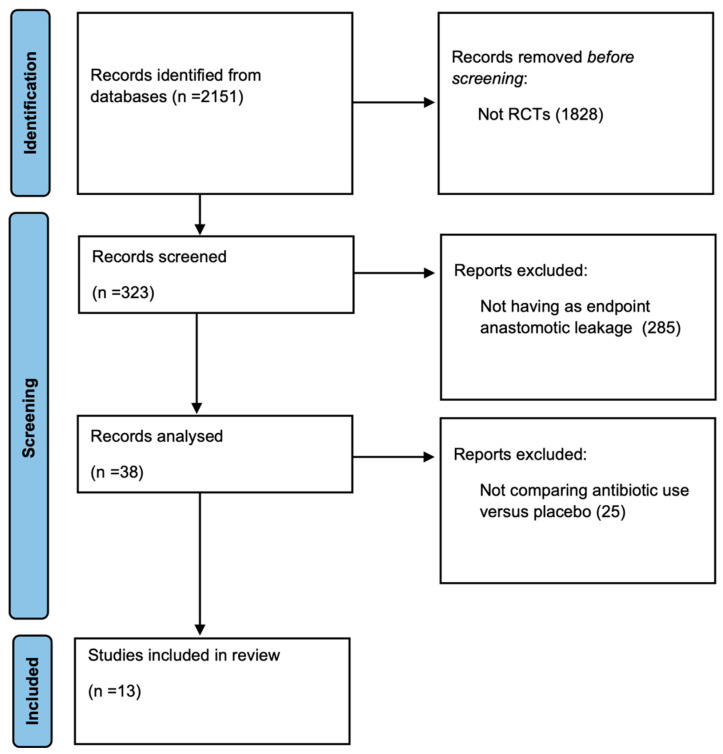
Flowchart of article selection.

**Figure 2 antibiotics-12-00397-f002:**
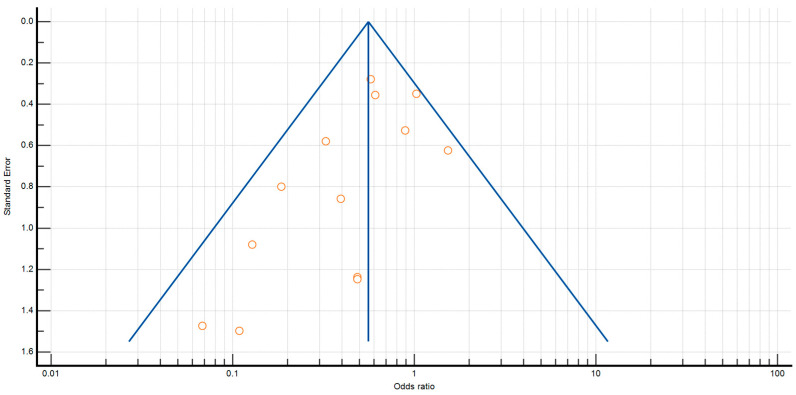
Funnel plot for anastomotic leaks.

**Figure 3 antibiotics-12-00397-f003:**
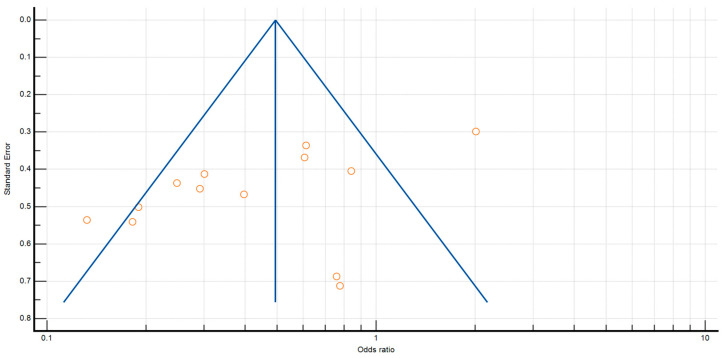
Funnel plot for surgical site infections.

**Figure 4 antibiotics-12-00397-f004:**
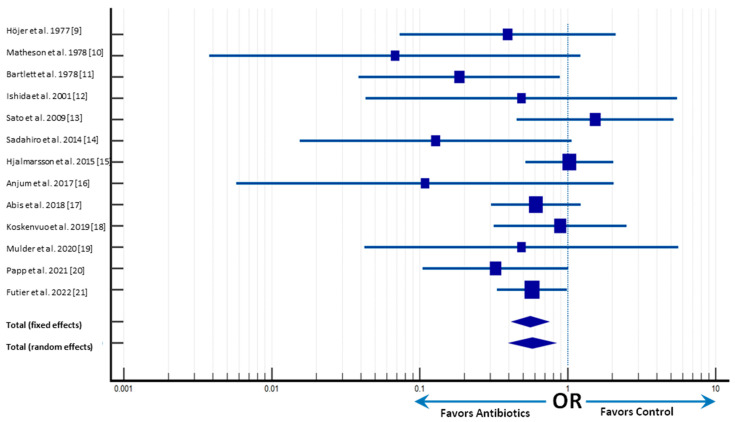
Forster plot of anastomotic leak rates. Odds ratio (OR) and 95% confidence intervals (CI) [9,10,11,12,13,14,15,16,17,18,19,20,21].

**Figure 5 antibiotics-12-00397-f005:**
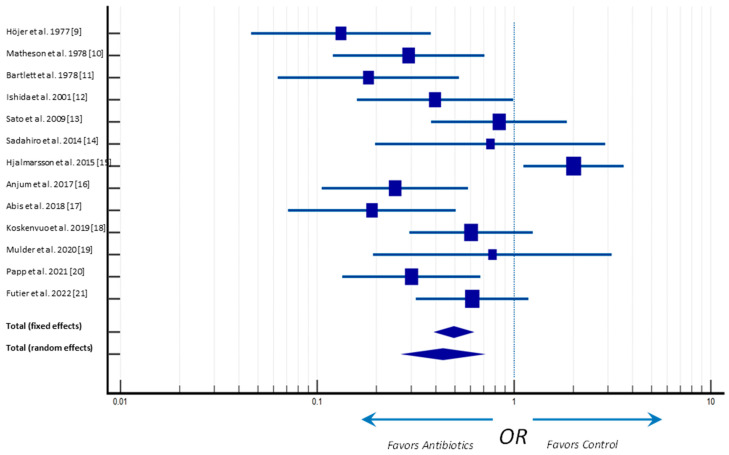
Forster plot of surgical site infections. Odds ratio (OR) and 95% confidence intervals (CI) [9,10,11,12,13,14,15,16,17,18,19,20,21].

**Table 2 antibiotics-12-00397-t002:** Version 2 of the Cochrane risk-of-bias tool for randomized trials to assess the risk of bias in the findings of each study. The tool is structured into five domains. Trials are judged for each domain as either having a ‘Low’/’Some concerns’/’High’ risk of bias with an overall judgement of the risk.

Author	1. Risk of Bias Arising from the Randomization Process	2.1 Risk of Bias Due to Deviations from the Intended Interventions (Effect of Assignment to Intervention)	2.2 Risk of Bias Due to Deviations from the Intended Interventions (Effect of Adhering to Intervention)	3. Risk of Bias Due to Missing Outcome Data	4. Risk of Bias in the Measurement of the Outcome	5. Risk of Bias in the Selection of the Reported Result	Overall Risk-of-Bias Judgement
Hojer et al. [9]	Low	Low	Low	Some concerns	Low	High	High
Matheson et al. [10]	Low	Low	Low	Some concerns	Low	High	High
Bartlett et al. [11]	Low	Low	Low	Some concerns	Low	Some concerns	Some concerns
Ishida et al. [12]	High	Low	High	Low	Low	High	High
Sato et al. [13]	Some concerns	Low	High	Low	Low	High	High
Sadahiro et al. [14]	Low	Low	Low	Some concerns	Low	High	High
Hjalmarsson et al. [15]	High	Low	High	Low	Low	High	High
Anjum et al. [16]	Low	Low	Low	Some concerns	Low	Low	Some concerns
Abis et al. [17]	High	Low	Some concerns	Low	Low	High	High
Koskenvuo et al. [18]	High	Low	High	Low	Low	High	High
Mulder et al. [19]	Low	Low	Low	Some concerns	Low	Some concerns	Some concerns
Papp et al. [20]	High	Some concerns	High	Low	Low	High	High
Futier et al. [21]	Low	Low	Low	Some concerns	Low	Some concerns	Some concerns

**Table 3 antibiotics-12-00397-t003:** Characterization of included RCTs.

Author	Year of Publication	Study Design	Number of Patients	Endpoints	Antibiotic Group	Control Group	Statistical Significance
Höjer et al. [9]	1977	Single-center, prospective, double-blinded, randomized trial	118		58	60	
				Anastomotic Leak	2	5	*p* < 0.001
				SSI	5	25	*p* < 0.001
Matheson et al. [10]	1978	Single-center, double-blinded, randomized controlled trial	110		51	59	
				Anastomotic Leak	0	7	*p* < 0.02
				SSI	9	25	*p* < 0.01
Bartlett et al. [11]	1978	Multicenter, double-blinded, randomized trial	116		56	60	
				Anastomotic Leak	2	10	*p* = 0.05
				SSI	5	21	*p* = 0.002
Ishida et al. [12]	2001	Single-center, single-blinded, randomized controlled trial	143		72	71	
				Anastomotic Leak	1	2	*p* = 0.050
				SSI	8	17	*p* = 0.035
Sato et al. [13]	2009	Multicenter, randomized trial	100		49	51	
				Anastomotic Leak	7	5	*p* = 0.8293
				SSI	20	23	*p* = 0.8293
Sadahiro et al. [14]	2014	Single-center, double-blinded, randomized trial	194		99	95	
				Anastomotic Leak	1	7	*p* = 0.014
				SSI	18	17	*p* = 0.004
Hjalmarsson et al. [15]	2015	Prospective, multicenter, single-blinded, randomized controlled trial	985		486	499	
				Anastomotic Leak	17	17	*p* = 0.95
				SSI	34	18	*p* = 0.022
Anjum et al. [16]	2017	Single-center, double-blinded, prospective, randomized trial	184		91	93	
				Anastomotic Leak	0	4	*p* = 0.004
				SSI	8	26	*p* = 0.001
Abis et al. [17]	2018	Superiority, open-label, multicenter, randomized trial	455		228	227	
				Anastomotic Leak	14	22	OR 0.61 (0.30–1.22)
				SSI	5	24	OR 0.19 (0.07–0.51)
Koskenvuo et al. [18]	2019	Multicenter, parallel, single-blinded randomized trial	396		196	200	
				Anastomotic Leak	7	8	CI 1.13 (0.40–3.16)
				SSI	13	21	CI 1.65 (0.80–3.40)
Mulder et al. [19]	2020	Multicenter, double-blind, placebo-controlled randomized trial	78		39	39	
				Anastomotic Leak	1	2	RR 0.50 (0.05–5.29)
				SSI	4	5	RR 0.80 (0.23–2.78)
Papp et al. [20]	2021	Multicentre, prospective, randomized, assessor-blinded trial	529		253	276	
				Anastomotic Leak	4	13	*p* = 0.020
				SSI	8	27	*p* = 0.001
Futier et al. [21]	2022	Multicenter, double-blinded, randomized, placebo-controlled trial	926		463	463	
				Anastomotic Leak	22	37	*p* = 0.046
				SSI	60	100	*p* = 0.001

**Table 4 antibiotics-12-00397-t004:** Routes of administration and regimens of antibiotic prophylaxis.

Author	Antibiotic	Administration Regimen	Administration Route
Hojer et al. [9]	Doxycycline 200 mg	- Started 4–6 h preop., single dose - Continued *o.d.* for 5 days postop.	Oral
Matheson et al. [10]	Neomycin 1 g and Metronidazole 200 mg	- Started 2 days preop. *t.d.s.*	Oral
Bartlett et al. [11]	Neomycin 1 g and Erythromycin 1 g	-Started 1 day preop. *t.d.s.*	Oral
Ishida et al. [12]	Metronidazole 400 mg and Kanamycin 500 mg	- Started 2 days preop. *b.d.* - Continued 3 days postop. *b.d.*	Oral
Sato et al. [13]	Cefotiam	- Started during skin incision - Continued 3 days postop. *t.d.s.*	Intravenous
Sadahiro et al. [14]	Kanamycin 0.5 g + Metronidazole 0.5 g	- Started 1 day preop. *t.d.s.*	Oral
Hjalmarsson et al. [15]	Sulfamethoxazole 800 mg/Trimethoprim 160 mg, and three tablets of Metronidazole 400 mg	-Started 2 h preop. single dose	Oral
Anjum et al. [16]	Metronidazole 400 mg and Levofloxacin 200 mg	- Started 1 day preop. *t.d.s.*	Oral
Abis et al. [17]	10 mL suspension containing 5 mL Amphotericin B 500 mg and 5 mL Colistin sulphate 100 mg and Tobramycin 80 mg	- Started 3 days preop. *q.i.d.* - Continued 3 days postop.	Oral
Koskenvuo et al. [18]	Neomycin 2 g and Metronidazole 2 g	- Started 1 day preop. *o.d.*	Oral
Mulder et al. [19]	Tobramycin 16 mg/mL and Colistin sulphate 20 mg/mL	- Started 3 days preop. *q.i.d.*	Oral
Papp et al. [20]	Ceftriaxone 2 g and Metronidazole 500 mg	-Started 1 day preop *t.d.s.*	Intravenous
Futier et al. [21]	Ornidazole 1 g	-Started 12 h preop. *o.d.*	Oral

(o.d: once daily; b.d: twice daily; t.d.s: three times daily; q.i.d: four times daily).

## Data Availability

The raw data is available upon reasonable request to the corresponding authors.

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
