# Peer review of "The Role of Antibiotic Prophylaxis in Anastomotic Leak Prevention during Elective Colorectal Surgery: Systematic Review and Meta-Analysis of Randomized Controlled Trials"

_antibiotics, 2023, doi:10.3390/antibiotics12020397_

Round 1
Reviewer 1 Report
The manuscript provides a systematic review and meta-analysis of previously published data emerging from randomized controlled trials, for a total of 13 studies, assessing the role of different regimens of antibiotic prophylaxis in preventing ALs after colorectal surgery. The results of this systematic review suggest that oral antibiotic bowel decontamination, whether used alone or in conjunction with mechanical bowel preparation, may have a considerable impact on the reduction of postoperative complications. According to the results of this review, the use of antibiotic prophylaxis prior to colorectal surgery should be strongly encouraged. The review is well conducted and well written; dealing with a delicate topic such as post-operative complications after colorectal surgery, the manuscript deserves publication.
Author Response
Reviewer 1
The manuscript provides a systematic review and meta-analysis of previously published data emerging from randomized controlled trials, for a total of 13 studies, assessing the role of different regimens of antibiotic prophylaxis in preventing ALs after colorectal surgery. The results of this systematic review suggest that oral antibiotic bowel decontamination, whether used alone or in conjunction with mechanical bowel preparation, may have a considerable impact on the reduction of postoperative complications. According to the results of this review, the use of antibiotic prophylaxis prior to colorectal surgery should be strongly encouraged. The review is well conducted and well written; dealing with a delicate topic such as post-operative complications after colorectal surgery, the manuscript deserves publication.
We would like to wholeheartedly thank Reviewer 1 for the kind comments.
Reviewer 2 Report
Dear Authors, thank you for allowing me to review your interesting systematic review on the usefulness of antibiotics associated to mechanical bowel preparation in colorectal surgery. The review is well done and scientifically sound. As it is clearly mentioned in the paper, unfortunately it does not provide any info on the best regimen and way of administration of antibiotic prophylaxis. Your paper deserves publication but I would be grateful if you could discuss this issue.
Author Response
Reviewer 2
Dear Authors, thank you for allowing me to review your interesting systematic review on the usefulness of antibiotics associated to mechanical bowel preparation in colorectal surgery. The review is well done and scientifically sound. As it is clearly mentioned in the paper, unfortunately it does not provide any info on the best regimen and way of administration of antibiotic prophylaxis. Your paper deserves publication but I would be grateful if you could discuss this issue.
We would like to thank Reviewer 2 for the kind comments and suggestions. Although our meta-analysis gave a clear indication on the effectiveness of oral antibiotic prophylaxis prior to colorectal surgery both in terms of anastomotic leaks and surgical site infections, due to the profoundly different antibiotic regimens used in the included RCTs it was not possible to find the best one. This has been further discussed in depth in the Discussion section (line 187-204).
Reviewer 3 Report
The authors conducted a systematic review and meta-analysis for assessing the effectivness of antibiotics preparation for elective colorectal surgery to prevent anastomotic leakage. They included 13 studies overall and concluded that antibiotic preparation is effective for preventing anastomotic leakage. Althougb previous cochrane review of on antibiotics prophylaxis in colorecta lsurgery concluded that anibiotics preparation is effective for preventing surgical site infection. (Nelson RL, Gladman E, Barbateskovic M. Antimicrobial
prophylaxis for colorectal surgery. Cochrane Database Syst Rev. 2014 May 9;2014(5):CD001181. doi:10.1002/14651858.CD001181.pub4. PMID: 24817514; PMCID: PMC8406790. ) However, they did not
investigate the anastomotic leakage. This systematic review and meta-analysis will give a new insight on antibiotics anaphylaixis in colorectal surgery. The authors listed the characteristics of included studies in detail and these information will be helpful to surgeons. However, I have some concerns for methodology.
i) The primary outcome of this SR&MA is anastomotic leakage as they stated in line 310-311. The conclusions of the main text is based on this primary outcome.However, their conclusion in abstract is "may have a considerable impact on the reduction of postoperative complications."Their conclusin is based on not only primary outcome but also surgical site infections (secondary outcome). The conclusions should be based one primary outcome. I suggest revising the conclusions of abstact so as to focus on primary outcome.
ii) I speculate the authors excluded the studies which did not investigate the anastomotic leakage according to Line 74-75. However, if the authors faithfully follow the method of Cochrane Handbook, they should have not excluded the studies for lack of outcomes. They should have included the studies and asked the authors of studies whether they investigated the anastomotic leakage or not. Besides, they had language restriction. There may have been more studies that can be included. The authors should refer to these shortages in
limitation section.
iii) In relation to ii), secondary outcomes are only considered for the literature included in this study. As the previous Cochrane review shows, there are plenty of studies which investigated SSI, yileding more than 200. The meta-analysis of only 13 studies for secondary outcome (SSI) is only confusing for readers. I suggest to remove the relevant description of SSI. If the authors want to investigate other outcomes than anastomotic leakage, I suggest to investigate adverse events.
iv) Cochrane RoB tool was used according to line 315-317. I did not find the results of assessment of RoB tool. The authors should present risk of bias summary and table. If the number of tables of figures exceeds the journal limit, they can be presented in supplementary materials.
There is only one minor concern;
Line 204-207 is a summary of Cochrane review, but reference #11 is not Cochrane review. Please make sure the reference is correct or no
Author Response
Reviewer 3
The authors conducted a systematic review and meta-analysis for assessing the effectivness of antibiotics preparation for elective colorectal surgery to prevent anastomotic leakage. They included 13 studies overall and concluded that antibiotic preparation is effective for preventing anastomotic leakage. Althougb previous cochrane review of on antibiotics prophylaxis in colorecta lsurgery concluded that anibiotics preparation is effective for preventing surgical site infection. (Nelson RL, Gladman E, Barbateskovic M. Antimicrobial prophylaxis for colorectal surgery. Cochrane Database Syst Rev. 2014 May 9;2014(5):CD001181. doi:10.1002/14651858.CD001181.pub4. PMID: 24817514; PMCID: PMC8406790. ) However, they did not investigate the anastomotic leakage. This systematic review and meta-analysis will give a new insight on antibiotics anaphylaixis in colorectal surgery. The authors listed the characteristics of included studies in detail and these information will be helpful to surgeons. However, I have some concerns for methodology.
We would like to thank Reviewer 3 for the kind comments and suggestions.
i) The primary outcome of this SR&MA is anastomotic leakage as they stated in line 310-311. The conclusions of the main text is based on this primary outcome.However, their conclusion in abstract is "may have a considerable impact on the reduction of postoperative complications."Their conclusin is based on not only primary outcome but also surgical site infections (secondary outcome). The conclusions should be based one primary outcome. I suggest revising the conclusions of abstact so as to focus on primary outcome.
The sentence has been changed according to the Reviewers’ suggestion.
ii) I speculate the authors excluded the studies which did not investigate the anastomotic leakage according to Line 74-75. However, if the authors faithfully follow the method of Cochrane Handbook, they should have not excluded the studies for lack of outcomes. They should have included the studies and asked the authors of studies whether they investigated the anastomotic leakage or not. Besides, they had language restriction. There may have been more studies that can be included. The authors should refer to these shortages in limitation section.
The Reviewer is correct regarding the fact that we excluded those studies not analyzing anastomotic leak rates and studies that were not reported in English. We screened all the randomized controlled trials for this outcome and if this was not present, supplementary material published for each study was additionally reviewed to evaluate whether this outcome was assessed even marginally.
As stated in the Cochrane Handbook, there are two reasons for incomplete (or missing) outcome data in clinical trials. Exclusions refer to situations in which some participants are omitted from reports of analyses, despite outcome data being available to the trialists. Within a published report those analyses with statistically significant differences between intervention groups are more likely to be reported than non-significant differences. This sort of ‘within-study publication bias’ is usually known as outcome reporting bias or selective reporting bias, and may be one of the most substantial biases affecting results from individual studies.
As we decided to include only randomized controlled trials in order to increase the power of this review and meta-analysis we decided to exclude those studies not reporting this outcome (either primary or secondary) as we considered this an incomplete outcome data bias as stated in the Cochrane Handbook. This has been added to the Limitations section.
iii) In relation to ii), secondary outcomes are only considered for the literature included in this study. As the previous Cochrane review shows, there are plenty of studies which investigated SSI, yileding more than 200. The meta-analysis of only 13 studies for secondary outcome (SSI) is only confusing for readers. I suggest to remove the relevant description of SSI. If the authors want to investigate other outcomes than anastomotic leakage, I suggest to investigate adverse events.
The Cochrane review included all trials evaluating surgical site infections. However, emergency surgical procedures were also included and these are known to have a greater risk of causing wound infections. Hence, we decided to exclude these procedures as we considered it as a major source of bias and included only elective cases.
Surgical site infections were included as a secondary outcome as some Authors seem to include anastomotic leaks in deep organ/space surgical site infections. Unfortunately, the definition of anastomotic leaks continues to be differing among reports. Hence, we included SSIs so as to not underestimate the rate of anastomotic insufficiency.
iv) Cochrane RoB tool was used according to line 315-317. I did not find the results of assessment of RoB tool. The authors should present risk of bias summary and table. If the number of tables of figures exceeds the journal limit, they can be presented in supplementary materials.
The table concerning the results deriving from the Cochrane RoB tool has been added to the manuscript (Table 2) and summarized in the results section (line 85-90).
There is only one minor concern;
Line 204-207 is a summary of Cochrane review, but reference #11 is not Cochrane review. Please make sure the reference is correct or no.
Please excuse this mistake. The reference was corrected.